# Hepatitis B Virus-Like Particle: Targeted Delivery of Plasmid Expressing Short Hairpin RNA for Silencing the *Bcl-2* Gene in Cervical Cancer Cells

**DOI:** 10.3390/ijms22052320

**Published:** 2021-02-26

**Authors:** Made Angga Akwiditya, Chean Yeah Yong, Mohd Termizi Yusof, Abdul Razak Mariatulqabtiah, Kok Lian Ho, Wen Siang Tan

**Affiliations:** 1Department of Microbiology, Faculty of Biotechnology and Biomolecular Sciences, Universiti Putra Malaysia, UPM Serdang, Selangor 43400, Malaysia; akwiditya@gmail.com (M.A.A.); yongcheanyeah@hotmail.com (C.Y.Y.); mohdtermizi@upm.edu.my (M.T.Y.); 2Department of Cell and Molecular Biology, Faculty of Biotechnology and Biomolecular Sciences, Universiti Putra Malaysia, UPM Serdang, Selangor 43400, Malaysia; mariatulqabtiah@upm.edu.my; 3Laboratory of Vaccine and Biomolecules, Institute of Bioscience, Universiti Putra Malaysia, UPM Serdang, Selangor 43400, Malaysia; 4Department of Pathology, Faculty of Medicine and Health Sciences, Universiti Putra Malaysia, UPM Serdang, Selangor 43400, Malaysia; klho@upm.edu.my

**Keywords:** short interference RNA, virus-like particle, anti-apoptotic *Bcl-2*, plasmid delivery, folic acid

## Abstract

Gene therapy research has advanced to clinical trials, but it is hampered by unstable nucleic acids packaged inside carriers and there is a lack of specificity towards targeted sites in the body. This study aims to address gene therapy limitations by encapsidating a plasmid synthesizing a short hairpin RNA (shRNA) that targets the anti-apoptotic *Bcl-2* gene using truncated hepatitis B core antigen (tHBcAg) virus-like particle (VLP). A shRNA sequence targeting anti-apoptotic *Bcl-2* was synthesized and cloned into the pSilencer 2.0-U6 vector. The recombinant plasmid, namely PshRNA, was encapsidated inside tHBcAg VLP and conjugated with folic acid (FA) to produce FA-tHBcAg-PshRNA VLP. Electron microscopy revealed that the FA-tHBcAg-PshRNA VLP has an icosahedral structure that is similar to the unmodified tHBcAg VLP. Delivery of FA-tHBcAg-PshRNA VLP into HeLa cells overexpressing the folate receptor significantly downregulated the expression of anti-apoptotic *Bcl-2* at 48 and 72 h post-transfection. The 3-(4, 5-dimethylthiazol-2-yl)-2, 5-diphenyltetrazolium bromide (MTT) assay demonstrated that the cells’ viability was significantly reduced from 89.46% at 24 h to 64.52% and 60.63%, respectively, at 48 and 72 h post-transfection. As a conclusion, tHBcAg VLP can be used as a carrier for a receptor-mediated targeted delivery of a therapeutic plasmid encoding shRNA for gene silencing in cancer cells.

## 1. Introduction

Cancers are notorious for killing millions of people in history. Until now, the disease shows no signs of plateauing, with approximately 20 million new cancer cases and 9 million related cancer deaths in 2018 [1]. Currently, the treatments employed to treat the disease include surgery, chemotherapy and radiation. The choice of treatments depends heavily on the location and severity of the cancers. However, these treatments have some limitations, for instance, surgery cannot completely remove tumors in unresectable areas, which increases the chance of recurrence. Besides, the side effects of chemotherapy and radiation can dramatically decrease the quality of patients’ lives [1,2]. Therefore, doctors and scientists have been searching for alternative treatments, including gene therapy.

Gene therapy research has advanced rapidly in the 20th century, and now it has reached clinical trials [3]. Generally, gene therapy functions by either replacing faulty genes with functional ones, suppressing the expression of harmful genes by using short interference RNA (siRNA), or introducing a new gene to prevent or fight against a disease with DNA vaccines [4,5]. The present study employed a plasmid DNA expressing a short hairpin RNA (shRNA) inside cancer cells. The shRNA is transcribed by RNA pol III in the nucleus, and transferred to the cytoplasm where a dicer converts the shRNA into siRNA and forms RNA-induced silencing complex (RISC), which binds and cleaves the target mRNA [6,7,8]. However, the delivery of siRNA, shRNA, and DNA molecules into cells remains a major challenge in clinical trials due to their poor cellular uptakes, and they are easily excreted by the kidneys [8,9]. An ideal delivery system should facilitate endosomal or lysosomal escape, have high transfection efficiency, specificity, and low toxicity [7,8,10]. To date, viral vectors such as adenovirus and retrovirus are used to deliver shRNA, but these viral vectors have some toxic effects on transfected cells [7,10]. Hence, nanoparticle-based delivery with high transfection, specificity, and low toxicity has been intensively studied to address the limitations of gene delivery [10,11]. The most commonly used nanoparticles are liposomal-, polymer-, metal- and protein-based nanoparticles [12]. While the liposomal-based nanoparticles have been approved for delivery of a cancer drug, doxorubicin [13,14], and these nanoparticles are less stable, rapidly removed by the immune system, and have unspecific delivery to healthy cells due to spontaneous membrane fusion [15,16]. The polymer-based nanoparticles do not have uniform shapes, and they are immunogenic and less stable [17]. However, the metal-based nanoparticles pose toxic effects to cells, and they lack target specificity [18]. Therefore, tremendous efforts have been made to produce more stable, homogeneous, less toxic, less immunogenic and highly specific nanoparticles. Among these nanoparticles, virus-like particles (VLPs) fulfill most of the criteria to be further developed as drug and gene delivery nano-vehicles.

VLPs are made of viral proteins, of which the particles formed and mimicked the conformation of authentic native viruses, but they lack the viral genomes [19]. One of the most extensively and intensively studied VLPs is the hepatitis B core antigen (HBcAg) VLP. In the present study, we used truncated HBcAg without the arginine rich C-terminal region (tHBcAg) VLP [20,21] to package a plasmid (3.2 kbp) expressing a shRNA that targets *B cell lymphoma-2* gene (*Bcl-2*).

In order to improve the targeting specificity of tHBcAg VLP, folic acid (FA) was conjugated to the primary amine groups exposed on the surface of the particle using the EDC (1-ethyl-3- [3-dimethylaminopropyl] carbodiimide) and sulfo-NHS (*N*-hydroxysulfosuccinimide) coupling method. As FA interacts specifically with the folate receptor (FR), which is highly expressed in many cancer cells, including cervical cancer cells [22], the tHBcAg VLP encapsidating the plasmid expressing shRNA (PshRNA) and conjugated with FA (namely: FA-tHBcAg-PshRNA) was therefore used to transfect HeLa cells (Scheme 1). Downregulation of anti-apoptotic *Bcl-2* expression and inhibition of HeLa cell proliferation demonstrated that FA-tHBcAg VLP can potentially be used to package shRNA vectors for targeted delivery to cancer cells overexpressing FR.

## 2. Results

### 2.1. Construction of Plasmid Carrying shRNA Sequence

Two oligonucleotides containing the sequences of *Bcl-2* siRNA sense, antisense, loop and RNA polymerase III terminator were synthesized (Figure 1A). These oligonucleotides were hybridized to form a double-stranded DNA containing *Bam*HI and *Hin*dIII restriction endonuclease sites at its 5′- and 3′-ends, respectively. The double-stranded DNA and pSilencer 2.0 hU6 vector were digested separately using *Bam*HI and *Hin*dIII, and ligated using T4 DNA ligase. Ligation mixture was introduced into *E. coli* TOP10 competent cells. The recombinant plasmid containing the insert (namely PshRNA) was extracted from a positive transformant. The insert in PshRNA was verified with DNA sequencing, and the result is shown in Figure 1B. The PshRNA expresses a short-hairpin RNA (shRNA) under the control of the human U6 promoter in mammalian cells. Dicer and RNA-induced silencing complex (RISC) proteins in the cells will then convert the shRNA to siRNA that silences the *Bcl-2* gene.

### 2.2. Encapsidation of PshRNA with tHBcAg VLP and Conjugation of VLP with FA

PshRNA was packaged inside tHBcAg VLP using the urea dissociation and association method. The VLP containing plasmid PshRNA (tHBcAg-PshRNA) was then conjugated with FA using the EDC and sulfo-NHS coupling method to produce FA-tHBcAg-PshRNA VLP. The plasmid-packaged VLPs (FA-tHBcAg-PshRNA and tHBcAg-PshRNA) and the control VLPs (tHBcAg, FA-tHBcAg) were then separated with sucrose density gradient ultracentrifugation, and protein concentration in each fraction was determined with the Bradford assay. Figure 2 shows that tHBcAg VLP and FA-tHBcAg VLP without PshRNA migrated into the sucrose gradient and formed a single peak (fractions 8 to 13). However, tHBcAg-PshRNA VLP and FA-tHBcAg-PshRNA VLP migrated into the gradient and formed two peaks; lighter density peak (fractions 8–12) and heavier density peak (fractions 13–18). The lighter density peak overlapped with the peak of control samples, suggesting the VLPs in the lighter density peak did not contain PshRNA. The heavier density peak was postulated to contain PshRNA-packaged VLPs. To confirm the postulation, the fractions of lighter density peak (fractions 8–12) and heavier density peak (fractions 13–18) were pooled separately, PshRNA was extracted using the alkaline lysis method, and the presence of the *Bcl*-2 sequence was confirmed with PCR. Agarose gel electrophoresis of the extracted plasmid and PCR product showed that the heavier density peak contained PshRNA carrying the *Bcl*-2 sequence, but the lighter density peak did not contain PshRNA (Figure 3). Collectively, the results showed that PshRNA was encapsidated by tHBcAg VLP, and migrated to the heavier density fractions in the sucrose gradient.

Spectrophotometry from wavelength 200 to 500 nm was employed to detect conjugated FA to tHBcAg VLPs. FA-tHBcAg VLP and FA-tHBcAg-PshRNA VLP gave rise to a higher absorbance at 360 nm, which corresponded well to the free FA (Figure 4), suggesting successful conjugation of FA to the tHBcAg VLPs. The samples without FA conjugation did not produce a higher absorbance at 360 nm.

### 2.3. Scanning Transmission Electron Microscopy (STEM) Analysis

STEM was performed to study the morphology of the tHBcAg VLP packaged with plasmid PshRNA and conjugated with FA. The result revealed that FA-tHBcAg-PshRNA formed icosahedral VLP similar to the control, tHBcAg VLP (Figure 5). This indicates that the packaging and conjugation processes did not affect the icosahedral structure of the VLPs.

### 2.4. Silencing of Bcl-2 Gene by FA-tHBcAg-PshRNA VLP

To study the silencing effect of FA-tHBcAg-PshRNA VLP on anti-apoptotic *Bcl-2* gene, the HeLa cell-line was chosen as it overexpresses FR. The HeLa cells were transfected separately with FA-tHBcAg VLP, plasmid PshRNA and FA-tHBcAg-PshRNA VLP. Non-transfected HeLa cells served as a negative control. The cells were incubated at 37 °C for 24, 48, and 72 h. Western blotting was performed to determine the downregulation of the *Bcl-2* gene, using *β-actin* as a housekeeping marker. The result showed that *Bcl-2* gene expression was significantly down-regulated at 48 and 72 h for the cells transfected with FA-tHBcAg-PshRNA, as compared to the untreated cells, or transfected with plasmid PshRNA and FA-tHBcAg VLP (Figure 6). This indicates that plasmid PshRNA was successfully delivered into HeLa cells by FA-tHBcAg VLP, and it cleaved the *Bcl-*2 mRNA and downregulated the expression of Bcl-2.

### 2.5. Viability of HeLa Cells Treated with FA-tHBcAg-PshRNA VLP

HeLa cells were separated into four groups in 96-well plates. The cells in each group were transfected separately with FA-tHBcAg VLP, plasmid PshRNA and FA-tHBcAg-PshRNA VLP, and an untreated group served as a negative control. Figure 7 shows that viable cells were significantly reduced in the group treated with FA-tHBcAg-PshRNA VLP at 48 h (64.52 ± 3.6%; *p <* 0.005) and at 72 h (60.63 ± 4.5%; *p <* 0.005) when compared to FA-tHBcAg and plasmid PshRNA treated groups. Interestingly, the cells that were treated with FA-tHBcAg-PshRNA VLP for 72 h had the lowest viability (60.63 ± 4.5%; *p <* 0.005), as compared to those treated with FA-tHBcAg (94.04 ± 3.4%) and PshRNA (90.79 ± 1.8%). This indicates that the plasmid PshRNA has been successfully delivered into the cells by FA-tHBcAg VLP, which subsequently down-regulated *Bcl-2*, and caused cell death.

## 3. Discussion

The main aim of this study was to package a plasmid expressing shRNA that targets *Bcl-2* using tHBcAg VLP, and to deliver the plasmid specifically to cancerous cells overexpressing FR. Different VLPs have been used as DNA carriers in several studies; *Macrobrachium rosenbergii* nodavirus VLP has been used to encapsidate Pie1-EGFP of 3.1 kbp [23], and hepatitis E virus VLP has been employed to encapsidate different plasmids, ranging from ~3 to ~11 kbp as DNA vaccines [24]. Another VLP that has been intensively studied is HBcAg VLP. The full-length HBcAg has an arginine rich region at its C-terminal region, which provides a positive internal surface for siRNA to be encapsidated inside the VLP by electrostatic interactions [25]. The VLP formed by the full-length HBcAg has been employed to package the RNA duplex (20 to 23 bp) [26,27,28]. Choi et al. [26] used the HBcAg VLP to deliver the red fluorescent protein siRNA to murine melanoma cells. In another study, Kong et al. [29] delivered Pokemon siRNA to hepatocellular carcinoma using the HBcAg VLP. These two studies employed genetic modifications to display an integrin-binding peptide (RGD) at the immunodominant region of HBcAg, located at the tip of the VLP spikes, for targeted delivery of the siRNAs to cancerous cells overexpressing cell membrane integrins. Additionally, Choi et al. [26] fused the p19 RNA binding protein at the C-terminal end of the HBcAg, which allows for the packaging of siRNA in the interior of the VLP. Both studies have successfully silenced the targeted genes and decreased the expression of mRNAs.

In the present study, we employed tHBcAg VLP to encapsidate a plasmid DNA, PshRNA, expressing shRNA. The monomer of tHBcAg (148 amino acid residues), is a *C*-terminally truncated mutant of the full-length HBcAg (183 residues) [21]. The VLP of tHBcAg is stable up to 80 °C [30,31], and from pH 4 to 10 [30]. In addition, the yield of tHBcAg produced in *E. coli* is five-folds higher than that of the full-length HBcAg [20]. Apart from the above-mentioned properties, the tHBcAg was employed because the *C*-terminal arginine rich region of the full-length HBcAg is partially exposed on the VLP, which causes unspecific uptake of the particle by cancerous cells via the Clathrin-mediated endocytosis [32,33]. Lee and Tan [34] demonstrated that the tHBcAg VLP can be used to package green fluorescent protein (GFP) in its interior cavity, which has paved the way for packaging different cargos, and their delivery to mammalian cells. Since then, the tHBcAg VLP has been employed widely for the delivery of fluorescein [35,36], oligonucleotide [35] and anticancer drugs [37,38,39,40]. In the present study, we demonstrated that tHBcAg VLP can be used to package plasmid PshRNA (~3.2 kbp) using the dissociation and association of the tHBcAg dimer reported by Lee and Tan [34]. The tHBcAg VLP encapsidating plasmid PshRNA can be separated and purified using sucrose density gradient ultracentrifugation.

tHBcAg VLP can be conjugated with different ligands, which include cell penetrating peptides and FA, to improve its specificity towards specific cancer cells [35,37,38,39,40]. FR is overexpressed in numerous cancer cell lines, including cervical cancer (HeLa), breast cancer (MCF7) and colorectal cancer (HT29) cells [41]. Therefore, in the present study, FA was conjugated to tHBcAg VLP for specific targeted delivery of PshRNA to HeLa cells. FA was conjugated by activating its carboxylate group using sulfo-NHS and EDC [39]. The activated carboxylate group reacts with the amine group of the Lys residues of tHBcAg. Spectrophotometry analysis revealed that FA was successfully conjugated to tHBcAg-PshRNA VLP. When observed under a scanning transmission electron microscope, the resulting FA-tHBcAg-PshRNA VLP has an icosahedral structure similar to that of the empty tHBcAg VLP, which indicates that the plasmid packaging and FA conjugation processes did not affect the VLP structure.

In this study, plasmid PshRNA was delivered into HeLa cells by FA-tHBcAg VLP. Silencing of the targeted gene, *Bcl-2*, begins with the transcription of shRNA from the human U6 RNA pol III promoter in the pSilencer 2.0 vector [42]. The shRNA is then cleaved into siRNA by a dicer in the cytoplasm, and the resulting siRNA initiates the formation of RISC, a complex that binds to the complementary mRNA and cleaves the targeted mRNA [8]. As a result, the expression of targeted gene (*Bcl-2*) is reduced. In this study, the reduction of *Bcl-2* expression was observed in western blots at 48 and 72 h post-transfection. A large body of literature reports that *Bcl-2* interacts with PUMA (p53 unregulated modulator of apoptosis), which, in turn, induces apoptosis by activating BAX (BCL-2-associated protein) or BAK (BCL-2-antagonist/killer) [42,43]. The activated BAX/BAK results in outer mitochondrial membrane permeabilization (MOMP), which releases apoptogenic factors that activate the caspase cascade [43,44]. Our study revealed that down-regulation of anti-apoptotic *Bcl-2* significantly reduced HeLa cell viability.

## 4. Materials and Methods

### 4.1. Construction of Plasmid Containing shRNA Sequence

Two oligonucleotides, namely forward insert (5′-GGGGGGGATCCGTAATAACG- TGCCTCATGAATTCAAGAGATTCATGAGGCACGTTATTATTTTTTGGAAAAGCTT- CCCCC-3′) and reverse insert (5′-GGGGGAAGCTTTTCCAAAAAATAATAACGTGCC- TCATGAATCTCTTGAATTCATGAGGCACGTTATTACGGATCCCCCCC-3′), which contain *Bcl-2* siRNA sequence ( [5′-TAATAACGTGCCTCATGAA-3′ (sense)] and [5′- TTCATGAGGCACGTTATTA-3′ (antisense)], respectively, were synthesized [44]. The oligonucleotides harbor *Bam*HI (5′-GGATCC-3′) and *Hin*dIII (5′-AAGCTT-3′) restriction endonuclease sites (underlined nucleotides). The oligonucleotides (1 µM in 100 µL) were hybridized by reducing the temperature from 94 °C to 30 °C, over ~2 h. The hybridized oligonucleotides (20 µg) were digested with *Bam*HI (20 U, Thermo Fisher Scientific, Waltham, MA, USA) and *Hin*dIII (20 U, Thermo Fisher Scientific, USA), and incubated at 37 °C overnight. The digested product was purified using the gel cleanup kit (Qiagen, Hilden, Germany).

Plasmid pAT206/Sept7shRNA was a kind gift from Matthew Krummel (Addgene plasmid #38299) [45]. It has a septin7 siRNA sequence inserted into the pSilencer 2.0-U6 vector [44]. This plasmid was digested with *Bam*HI and *Hin*dIII for 30 min at 37 °C to remove the septin7 siRNA. The linearized pSilencer 2.0-U6 vector was purified using the gel cleanup kit by following the manufacturer’s protocol (Qiagen, Germany).

The digested vector (123 ng) and insert (130 ng) were ligated using T4 DNA ligase (3 U; Thermo Fisher Scientific, USA) in a 30 µL reaction, at 22 °C for 1 h, and continued incubation at 4 °C overnight. The ligation mixture (5 µL) was introduced into *E. coli* TOP10 competent cells using the heat shock method. The positive transformant containing recombinant plasmid, PshRNA, was verified using PCR. The nucleotide sequence of the recombinant plasmid was determined using the dideoxy-chain termination method [46].

### 4.2. Production and Purification of Truncated Hepatitis B Core Antigen (tHBcAg)

tHBcAg (residues 1–148) was produced in the *E. coli* strain W3110IQ harboring pR1-11E plasmid [20]. The bacteria were grown in Luria Bertani (LB) broth containing ampicillin (100 µg/mL), and tHBcAg production was induced with IPTG (0.25 mM in 500 mL culture) as described by Tan et al. [20]. The cells were harvested by centrifugation at 8000× *g* for 30 min at 4 °C. The cell pellet was resuspended with Tris-Triton buffer [50 mM Tris (pH 8.0), 0.1% (*v*/*v*) Triton X-100], and sonicated at 200 Hz for 5 min. After sonication, the cells were centrifuged at 14,000× *g* for 20 min at 4 °C. Ammonium sulphate [35% (*w*/*v*) saturation] was used to precipitate the proteins in the supernatant, and the precipitated proteins were collected by centrifugation (14,000× *g* for 20 min at 4 °C). The pellet was dissolved in Tris-NaCl buffer [50 mM Tris (pH 8.0), 100 mM NaCl], and dialyzed in 1.5 L of the same buffer at 4 °C for 5 h. The dialysis was continued overnight using a new batch of fresh dialysis buffer. The dialyzed sample was separated using 8–40% (*w*/*v*) sucrose density gradient ultracentrifugation (SW 41Ti rotor, Beckman Coulter, Brea, CA, USA) at 210,000× *g* for 5 h at 4 °C. Then, the sucrose gradient was fractionated, and analyzed using sodium dodecyl sulfate-polyacrylamide gel electrophoresis (SDS-PAGE). Fractions that contained tHBcAg were pooled, and the Bradford assay was used to determine its concentration [47]. The formation of VLP was determined using scanning transmission electron microscopy (STEM).

### 4.3. Encapsidation of Plasmid PshRNA with tHBcAg VLP

The encapsidation of plasmid PshRNA was conducted according to Lee & Tan [34]. Urea (2.5 M) was used to dissociate tHBcAg (200 µg) at room temperature for 3 h with gentle shaking. PshRNA (20 µg) was added into the dissociated tHBcAg. The mixture was dialyzed against Tris-NaCl buffer to re-associate tHBcAg, and concentrated using the Vivaspin concentrator (30 kDa cut off, Sartorius, Göttingen, Germany). The concentrated sample was conjugated with FA.

### 4.4. Conjugation of Folic Acid (FA) to tHBcAg VLP

Conjugation of FA to tHBcAg VLP was performed according to Biabanikhankahdani et al. [37]. FA (5 mg; Sigma-Aldrich, St. Louis, MO, USA) was mixed with EDC (20 mg) and sulfo-NHS (20 mg) and incubated at room temperature for 4 h. The tHBcAg VLP and tHBcAg-PshRNA VLP were added separately with FA in 2:1 ratio. The mixtures were incubated with gentle agitation at 4 °C for 8–16 h, before loading onto sucrose gradients (8–40%) and centrifuged at 210,000× *g*, for 5 h at 4 °C, to remove free plasmids and unconjugated FA. UV absorbance (200–500 nm) of FA-conjugated tHBcAg (FA-tHBcAg), FA-conjugated tHBcAg VLP encapsidating PshRNA (FA-tHBcAg-PshRNA VLP), tHBcAg VLP encapsidating PshRNA (tHBcAg-PshRNA), tHBcAg without going through the packaging and FA conjugation processes (tHBcAg), FA, and PshRNA was measured using a spectrophotometer (Jenway, Staffordshire, UK).

### 4.5. Density Analysis of tHBcAg VLP Encapsidating PshRNA and Conjugated with FA

FA-tHBcAg-PshRNA, FA-tHBcAg, tHBcAg-PshRNA and tHBcAg samples were separated with sucrose density gradient ultracentrifugation as described above. Sucrose gradients were fractioned (0.5 mL per fraction), and the protein concentration in each fraction was determined using the Bradford assay [47]. The peaks containing VLPs were pooled, and plasmid extraction was performed using the alkaline lysis method [48]. The presence of shRNA sequence in the plasmid was determined using PCR containing dNTPs (200 µM), forward primer (5′-GTTGGGTAACGCCAGGGTTTTC-3′); reverse primer (5′-CCGTAACTTGAAAGTATTTCGAT-3′), *Taq Polymerase* (0.026 U/µL) and PshRNA (60–100 ng). PCR profile was initial denaturation at 94 °C for 5 min, 30 cycles of denaturation at 94 °C for 45 s, annealing at 55 °C for 45 s, extension at 72 °C for 45 s, and a final extension at 72 °C for 10 min.

### 4.6. Scanning Transmission Electron Microscopy (STEM)

FA-tHBcAg VLP, FA-tHBcAg-PshRNA VLP and tHBcAg VLP samples (0.35 mg/mL; 15 µL) were absorbed onto carbon-coated grids (300 mesh) for 10 min. The grids were stained using 2% (*w*/*v*) uranyl acetate (Sigma-Aldrich, USA) for 10 min. The samples were observed under a scanning transmission electron microscope (Hitachi su8230, Ibaraki, Japan).

### 4.7. Delivery of Plasmid PshRNA into HeLa Cells Using tHBcAg VLP Conjugated with FA

HeLa cells were obtained from American Type Culture Collection (ATCC), and cultured in DMEM (Gibco, Schwalbach, Germany) supplemented with 10% FBS (Sigma, Germany) at 37 °C in 5% CO_2_. The cells were seeded in 6-well plates, and incubated for 24 h at 37 °C until they reached 70% confluency. The culture medium was aspirated, and a fresh medium (1 mL) was added into the wells. FA-tHBcAg-PshRNA VLP (200 µg in 100 µL medium), FA-tHBcAg VLP (200 µg in 100 µL medium), and PshRNA (20 µg in 100 µL medium) samples were added separately to the cells and incubated further at 37 °C in 5% CO_2_ for 24, 48, and 72 h. The non-transfected cells, and the cells added with FA-tHBcAg VLP and PshRNA (naked plasmid) served as negative controls.

### 4.8. SDS-PAGE and Western Blotting

HeLa cells were divided into four different treatment groups (non-transfected, transfected with FA-tHBcAg VLP, transfected with FA-tHBcAg-PshRNA VLP and transfected with PshRNA). The cells were harvested after 24, 48 and 72 h incubation by trypsinization, and centrifuged at 14,000× *g* for 10 min at 4 °C. The pellet was dissolved in water, and mixed with 6× SDS loading buffer [375 mM Tris-HCl (pH 6.8), 6% (*w*/*v*) SDS, 4.8% (*v*/*v*) glycerol, 0.03% (*w*/*v*) bromophenol blue, 9% (*v*/*v*) β-mercaptoethanol], boiled for 10 min, and electrophoresed (16 mA, 1 h) in SDS-15% polyacrylamide gels. Separated proteins in the gels were electro-transferred (24 V, 35 min) onto nitrocellulose membranes using a semi-dry blotting system. The membranes were blocked with skim milk [Anlene, Malaysia, 10% (*w*/*v)* in TBS (50 mM Tris-HCl, 150 mM NaCl; pH 7.4)] for 2 h at room temperature. The membranes were washed three times (5 min each) with TBST [TBS containing 0.01% (*v*/*v)* Tween-20] before anti-*Bcl-*2 monoclonal antibody [*Bcl-2* (C-2): sc-7382, 1:500 dilution in TBS, Santa Cruz Biotechnology, Germany] or anti-*β-actin* monoclonal antibody [*β*-*actin* (C4): sc-47778, 1:1000 dilution in TBS, Santa Cruz Biotechnology, Germany] was incubated with the membranes and incubated at 4 °C overnight. The membranes were then washed three times with TBST as mentioned above, and the secondary anti-mouse antibody (1:5000 dilution in TBS, Seracare Life Sciences, Milford, MA, USA) was added and incubated for 1 h at room temperature. Color development was done by adding the alkaline phosphate buffer (100 mM Tris-HCl, 100 mM NaCl, 5 mM MgCl_2_, pH 9.5) containing 5-bromo-4-chloro-3-indolyl phosphate (BCIP; 3.5 mM) and nitro blue tetrazolium (NBT; 3.7 mM) for 10 min, and the color development was stopped using distilled water.

### 4.9. Cell Viability Assay

The 3-(4, 5-dimethylthiazol-2-yl)-2, 5-diphenyltetrazolium bromide (MTT) assay was used to determine cell viability. HeLa cells (10^4^ cells per well) were seeded into 96-well plates, and incubated at 37 °C in 5% CO_2_ for 24 h. FA-tHBcAg-PshRNA VLP, FA-tHBcAg VLP, and plasmid PshRNA were added to the cells, and incubated for 24, 48, and 72 h at 37 °C in 5% CO_2_. Non-transfected HeLa cells served as a negative control. Then, the media (50 µL) were removed and MTT reagent (20 µL) was added. The mixture was incubated at 37 °C in 5% CO_2_ for 3.5 h. Then, dimethyl sulfoxide (150 µL) was added and shaken gently for 15 min. Absorbance was read at 570 nm using a microplate reader (BioTek instruments, Winooski, VT, USA).

### 4.10. Statistical Analysis

Statistical analysis was performed using the GraphPad Prism statistics software (version 9.0.0). A *p* value less than 0.01 is considered statistically significant.

## 5. Conclusions

Plasmid PshRNA expressing shRNA that targets the *Bcl-2* gene was successfully constructed and encapsidated inside tHBcAg VLP. The tHBcAg-PshRNA was conjugated with FA to target HeLa cells overexpressing FR. The structure of tHBcAg VLP was not affected by the plasmid packaging and FA conjugation processes. The FA-tHBcAg VLP successfully delivered PshRNA into HeLa cells, which silenced *Bcl-2* mRNA and reduced the cell viability. This study demonstrated that tHBcAg VLP displaying FA is a potential carrier for the specific delivery of therapeutic plasmid DNA into cancer cells overexpressing FR.

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
