# Peer review of "Hepatitis B Virus-Like Particle: Targeted Delivery of Plasmid Expressing Short Hairpin RNA for Silencing the Bcl-2 Gene in Cervical Cancer Cells"

_ijms, 2021, doi:10.3390/ijms22052320_

Round 1

Reviewer 1 Report

In the present manuscript “Hepatitis B virus-like particle: targeted delivery of plasmid  expressing short hairpin RNA for silencing the Bcl-2 gene in  cervical cancer cells” a truncated HBcAg form, without the arginine rich C-terminal region 65 (tHBcAg) VLP  was used to package a plasmid expressing a short hairpin RNA (shRNA) that targets Bcl-2 gene. A further improvement of the targeting specificity of tHBcAg VLP was obtained with the conjugation with folic acid (FA) that interacts specifically with folate receptor (FR), which is highly expressed in many cancer cells including cervical cancer cells. The present research was finalized by  infecting HeLa cells and testing the Bcl-2 gene silencing and the cell viability.

The paper is well written and provides a very important clinical perspective for the treatment of cancer cells in a targeted and effective context.  However, there are few issues that deserve revision and comments.

1- In in vitro experiments, the HeLa cells were transfected with FA-tHBcAg VLP, plasmid PshRNA and FA-tHBcAg-PshRNA VLP for 24, 48, and 72 hours. The transfection protocol is unclear, was a transfection agent used? Was the culture medium changed for transfection? Authors should provide full details of the experimental procedure in the materials and methods paragraph.

2- Transfection of HeLa cells with PshRNA plasmid should induce a silencing of the Bcl-2 gene and a decrease in protein expression, but this is not observable from the WB. Does PshRNA plasmid express an unspecific siRNA? Can the authors comment on this aspect and provide an explanation?

Author Response

REVIEWER 1

Comment: In in vitro experiments, the HeLa cells were transfected with FA-tHBcAg VLP, plasmid PshRNA and FA-tHBcAg-PshRNA VLP for 24, 48, and 72 hours. The transfection protocol is unclear, was a transfection agent used? Was the culture medium changed for transfection? Authors should provide full details of the experimental procedure in the materials and methods paragraph.

Response: Many thanks for the comment. We have now provided the full details for the delivery of plasmid PshRNA into HeLa cells using tHBcAg VLP conjugated with FA, in the methodology section (page 12, lines 372-377). The main aim of this study was to deliver plasmid PshRNA into the cells using tHBcAg VLP conjugated with folic acid (FA), therefore no transfection agent was used to deliver the FA-tHBcAg-PshRNA VLP, FA-tHBcAg VLP, and PshRNA. The PshRNA was added into culture as a naked plasmid DNA, which served as a negative control. FA-tHBcAg-PshRNA VLP stands for the PshRNA packaged inside tHBcAg VLP and conjugated with folic acid.  FA-tHBcAg VLP indicates tHBcAg VLP conjugated with folic acid, which served as a negative control too.

Comment: Transfection of HeLa cells with PshRNA plasmid should induce a silencing of the Bcl-2 gene and a decrease in protein expression, but this is not observable from the WB. Does PshRNA plasmid express an unspecific siRNA? Can the authors comment on this aspect and provide an explanation?

Response: Many thanks for the comment. The PshRNA was added as a naked plasmid DNA, which served as a negative control. This plasmid was not packaged with tHBcAg VLP conjugated with folic acid, thus it cannot enter the HeLa cells and did not express any siRNA for silencing the Bcl-2 gene and the protein expression.

Reviewer 2 Report

The manuscript titled “Hepatitis B virus-like particle: targeted delivery of plasmid 2 expressing short hairpin RNA for silencing the Bcl-2 gene in 3 cervical cancer cells” is well written and presented by the authors. After critical review of the manuscript, the paper can be accepted for publication after minor revisions provided below.

Introduction: Please provide rationale and information on the mechanism of “plasmid 2 expressing short hairpin RNA” in silencing Bcl-2 gene and also provide existing approaches for delivery of shRNA and its challenges.

It could have been advantageous if the authors conducted cellular uptake studies (imaging/FACs) to compare between folic acid conjugated VLPs and unconjugated VLPs (this is not mandatory).

Overall, the investigation looks interesting and could significantly contribute to the advancement of gene therapy.

Author Response

REVIEWER 2

Comment: Introduction: Please provide rationale and information on the mechanism of “plasmid 2 expressing short hairpin RNA” in silencing Bcl-2 gene and also provide existing approaches for delivery of shRNA and its challenges.

Response: Many thanks for the comment. The present study employed a plasmid DNA expressing a short hairpin RNA (shRNA) inside cancer cells. The shRNA is transcribed by RNA pol III in the nucleus, and transferred to the cytoplasm where a dicer converts the shRNA into siRNA and forms RNA-induced silencing complex (RISC), which binds and cleaves the target mRNA (Kunkel and Pederson, 1989; Sliva and Schnierle, 2010; Zhou et al., 2014). However, the delivery of siRNA, shRNA, and DNA molecules into cells remains a major challenge in clinical trials due to their poor cellular uptakes, and they are easily excreted by the kidneys (Zhou et al., 2014; Lowe et al., 2006). An ideal delivery system should facilitate endosomal or lysosomal escape, have high transfection efficiency, specificity, and low toxicity (Sliva and Schnierle., 2010; Zhou et al., 2014; Burnett et al., 2011). To date, viral vectors such as adenovirus and retrovirus are used to deliver shRNA, but these viral vectors have some toxic effects on transfected cells (Sliva and Schnierle., 2010; Burnett et al., 2011). This additional information has now been added to the introduction section (Page 2, lines 48-57). The references have been added accordingly.

Comment: It could have been advantageous if the authors conducted cellular uptake studies (imaging/FACs) to compare between folic acid conjugated VLPs and unconjugated VLPs (this is not mandatory).

Response: We kindly appreciate your comment. We are constrained by the government-imposed lockdown to curb the spread of COVID-19. We are not allowed to enter our laboratory, and unsure when the lockdown will be lifted to enable us to provide this information. Kindly accept our apologies. However in our previous study using live cell imaging, Biabanikhankahdani et al. (2016) demonstrated that tHBcAg VLP conjugated with folic acid and packaged with doxorubicin was able to deliver the drug into colorectal cancer cells, while the unconjugated tHBcAg VLP packaging doxorubicin was not able to deliver the drug into the cells. 

Reference:

Biabanikhankahdani, R.; Alitheen, N.B.M.; Ho, K.L.; Tan, W.S. pH-responsive virus-like nanoparticles with enhanced tumour-targeting ligands for cancer drug delivery. Sci. Rep. 2016, 6, 1–13, doi:10.1038/srep37891.